# Learning from uncertain curves:
# The $2$-Wasserstein metric for Gaussian processes

**Anton Mallasto**
Department of Computer Science
University of Copenhagen
mallasto@di.ku.dk

**Aasa Feragen**
Department of Computer Science
University of Copenhagen
aasa@di.ku.dk

## Abstract

We introduce a novel framework for statistical analysis of populations of non-degenerate Gaussian processes (GPs), which are natural representations of uncertain curves. This allows inherent variation or uncertainty in function-valued data to be properly incorporated in the population analysis. Using the 2-Wasserstein metric we geometrize the space of GPs with $L^2$ mean and covariance functions over compact index spaces. We prove uniqueness of the barycenter of a population of GPs, as well as convergence of the metric and the barycenter of their finite-dimensional counterparts. This justifies practical computations. Finally, we demonstrate our framework through experimental validation on GP datasets representing brain connectivity and climate development. A MATLAB library for relevant computations will be published at https://sites.google.com/view/antonmallasto/software.

## 1 Introduction

*Gaussian processes* (GPs, see Fig. 1) are the counterparts of Gaussian distributions (GDs) over functions, making GPs natural objects to model uncertainty in estimated functions. With the rise of GP modelling and probabilistic numerics, GPs are increasingly used to model uncertainty in function-valued data such as segmentation boundaries [17, 19, 30], image registration [38] or time series [28]. Centered GPs, or covariance operators, appear as image features in computer vision [12,16,25,26] and as features of phonetic language structure [23]. A natural next step is therefore to analyze populations of GPs, where performance depends crucially on proper incorporation of inherent uncertainty or variation. This paper contributes a principled framework for population analysis of GPs based on Wasserstein, a.k.a. earth mover's, distances.

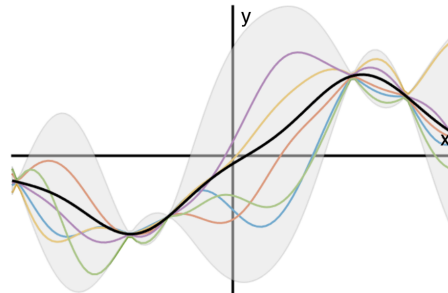

Figure 1: An illustration of a GP, with mean function (*in black*) and confidence bound (*in grey*). The colorful curves are sample paths of this GP.

The importance of incorporating uncertainty into population analysis is emphasized by the example in Fig. 2, where each data point is a GP representing the minimal temperature in the Siberian city Vanavara over the course of one year [9, 34]. A naïve way to compute its average temperature curve is to compute the per-day mean and standard deviation of the yearly GP mean curves. This is shown in the bottom right plot, and it is clear that the temperature variation is grossly underestimated, especially in the summer season. The top right figure shows the mean GP obtained with our proposed framework, which preserves a far more accurate representation of the natural temperature variation.

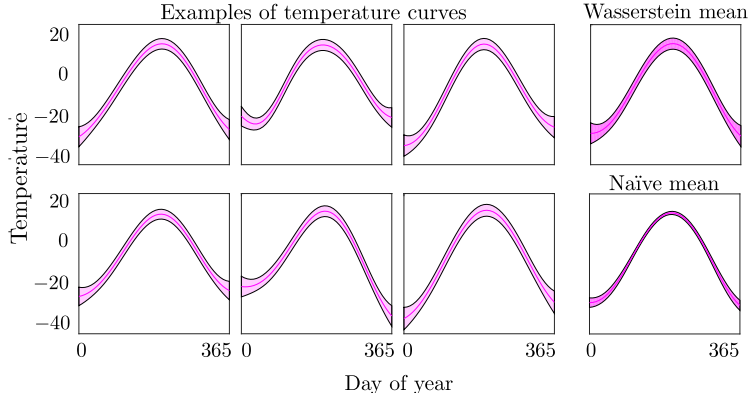

Figure 2: **Left:** Example GPs describing the daily minimum temperatures in a Siberian city (see Sec. 4). **Right top:** The mean GP temperature curve, computed as a Wasserstein barycenter. Note that the inherent variability in the daily temperature is realistically preserved, in contrast with the naïve approach. **Right bottom:** A naïve estimation of the mean and standard deviation of the daily temperature, obtained by taking the day-by-day mean and standard deviation of the temperature. All figures show a 95% confidence interval.

We propose analyzing populations of GPs by geometrizing the space of GPs through the *Wasserstein distance*, which yields a metric between probability measures with rich geometric properties. **We contribute** i) closed-form solutions for arbitrarily good approximation of the Wasserstein distance by showing that the 2-Wasserstein distance between two finite-dimensional GP representations converges to the 2-Wasserstein distance of the two GPs; and ii) a characterization of a non-degenerate barycenter of a population of GPs, and a proof that such a barycenter is unique, and can be approximated by its finite-dimensional counterpart.

We evaluate the Wasserstein distance in two applications. First, we illustrate the use of the Wasserstein distance for processing of uncertain white-matter trajectories in the brain segmented from noisy diffusion-weighted imaging (DWI) data using *tractography*. It is well known that the noise level and the low resolution of DWI images result in unreliable trajectories (*tracts*) [24]. This is problematic as the estimated tracts are e.g. used for surgical planning [8]. Recent work [17, 30] utilizes probabilistic numerics [29] to return *uncertain* tracts represented as GPs. We utilize the Wassertein distance to incorporate the estimated uncertainty into typical DWI analysis tools such as tract clustering [37] and visualization. Our second study quantifies recent climate development based on data from Russian meteorological stations using permutation testing on population barycenters, and supplies interpretability of the climate development using GP-valued kernel regression.

**Related work.** Multiple frameworks exist for comparing Gaussian distributions (GDs) represented by their covariance matrices, including the Frobenius, Fisher-Rao (affine-invariant), log-Euclidean and Wasserstein metrics. Particularly relevant to our work is the 2-Wasserstein metric on GDs, whose Riemannian geometry is studied in [33], and whose barycenters are well understood [1, 4].

A body of work exists on generalizing the aforementioned metrics to the infinite-dimensional covariance operators. As pointed out in [23], extending the affine-invariant and Log-Euclidean metrics is problematic as covariance operators are not compatible with logarithmic maps and their inverses are unbounded. These problems are avoided in [25, 26] by regularizing the covariance operators, but unfortunately, this also alters the data in a non-unique way. The Procrustes metric from [23] avoids this, but as it is, only defines a metric between covariance operators.

The 2-Wasserstein metric, on the other hand, generalizes naturally from GDs to GPs, does not require regularization, and can be arbitrarily well approximated by a closed form expression, making the computations cheap. Moreover, the theory of optimal transport [5, 6, 36] shows that the Wasserstein metric yields a rich geometry, which is further demonstrated by the previous work on GDs [33].

After this work was presented in NIPS, a preprint appeared [20] which also studies convergence results and barycenters of GPs in the Wasserstein geometry, in a more general setting.

**Structure.** Prior to introducing the Wasserstein distance between GPs, we review GPs, their Hilbert space covariance operators and the corresponding Gaussian measures in Sec. 2. In Sec. 3 we introduce the Wasserstein metric and its barycenters for GPs and prove convergence properties of the metric and barycenters, when GPs are approximated by finite-dimensional GDs. Experimental validation is found in Sec. 4, followed by discussion and conclusion in Sec. 5.

## 2   Prerequisites

**Gaussian processes and measures.** A *Gaussian process* (GP) $f$ is a collection of random variables, such that any finite restriction of its values $(f(x_i))_{i=1}^N$ has a joint Gaussian distribution, where $x_i \in X$, and $X$ is the *index set*. A GP is entirely characterized by the pair

$$m(x) = \mathbb{E}\left[f(x)\right], \ k(x, x') = \mathbb{E}\left[(f(x) - m(x))(f(x') - m(x'))\right] , \tag{1}$$

where $m$ and $k$ are called the *mean function* and *covariance function*, respectively. We use the notation $f \sim \mathcal{GP}(m, k)$ for a GP $f$ with mean function $m$ and covariance function $k$. It follows from the definition that the covariance function $k$ is symmetric and positive semidefinite. We say that $f$ is *non-degenerate*, if $k$ is strictly positive definite. We will assume the GPs used to be non-degenerate.

GPs relate closely to *Gaussian measures* on Hilbert spaces. Given probability spaces $(X, \Sigma_X, \mu)$ and $(Y, \Sigma_Y, \nu)$, we say that the measure $\nu$ is a *push-forward* of $\mu$ if $\nu(A) = \mu(T^{-1}(A))$ for a measurable $T \colon X \to Y$ and any $A \in \Sigma_Y$. Denote this by $T_\#\mu = \nu$. A Borel measure $\mu$ on a separable Hilbert space $\mathcal{H}$ is a *Gaussian measure*, if its push-forward with respect to any non-zero continuous element of the dual space of $\mathcal{H}$ is a non-degenerate Gaussian measure on $\mathbb{R}$ (i.e., the push-forward gives a univariate Gaussian distribution). A Borel-measurable set $B$ is a *Gaussian null set*, if $\mu(B) = 0$ for any Gaussian measure $\mu$ on $X$. A measure $\nu$ on $\mathcal{H}$ is *regular* if $\nu(B) = 0$ for any Gaussian null set $B$. Note that regular Gaussian measures correspond to non-degenerate GPs.

**Covariance operators.** Denote by $L^2(X)$ the space of $L^2$-integrable functions from $X$ to $\mathbb{R}$. The covariance function $k$ has an associated integral operator $K \colon L^2(X) \to L^2(X)$ defined by

$$[K\phi](x) = \int_X k(x, s)\phi(s)ds, \ \forall \phi \in L^2(X) , \tag{2}$$

called the *covariance operator* associated with $k$. As a by-product of the 2-Wasserstein metric on centered GPs, we get a metric on covariance operators. The operator $K$ is Hilbert-Schmidt, self-adjoint, compact, positive, and of trace class, and the space of such covariance operators is a convex space. Furthermore, the assignment $k \mapsto K$ from $L^2(X \times X)$ is an isometric isomorphism onto the space of Hilbert-Schmidt operators on $L^2(X)$ [7, Prop. 2.8.6]. This justifies us to write both $f \sim \mathcal{GP}(m, K)$ and $f \sim \mathcal{GP}(m, k)$.

**Trace of an operator.** The Wasserstein distance between GPs admits an analytical formula using traces of their covariance operators, as we will see below. Let $(\mathcal{H}, \langle \cdot, \cdot \rangle)$ be a *separable* Hilbert space with the orthonormal basis $\{e_k\}_{k=1}^\infty$. Then the *trace* of a bounded linear operator $T$ on $\mathcal{H}$ is given by

$$\operatorname{Tr} T := \sum_{k=1}^\infty \langle Te_k, e_k \rangle , \tag{3}$$

which is absolutely convergent and independent of the choice of the basis if $\operatorname{Tr} (T^*T)^{\frac{1}{2}} < \infty$, where $T^*$ denotes the adjoint operator of $T$ and $T^{\frac{1}{2}}$ is the square-root of $T$. In this case $T$ is called a *trace class operator*. For positive self-adjoint operators, the trace is the sum of their eigenvalues.

**The Wasserstein metric.** The *Wasserstein metric* on probability measures derives from the optimal transport problem introduced by Monge and made rigorous by Kantorovich. The $p$-Wasserstein distance describes the minimal cost of transporting the unit mass of one probability measure into the unit mass of another probability measure, when the cost is given by a $L^p$ distance [5, 6, 36].

Let $(M, d)$ be a Polish space (complete and separable metric space) and denote by $\mathcal{P}_p(M)$ the set of all probability measures $\mu$ on $M$ satisfying $\int_M d^p(x, x_0)d\mu(x) < \infty$ for some $x_0 \in M$. The $p$-Wasserstein distance between two probability measures $\mu, \nu \in \mathcal{P}_p(M)$ is given by

$$W_p(\mu, \nu) = \left( \inf_{\gamma \in \Gamma[\mu, \nu]} \int_{M \times M} d^p(x_1, x_2)d\gamma(x_1, x_2) \right)^{\frac{1}{p}}, \ (x_1, x_2) \in M \times M, \tag{4}$$

where $\Gamma[\mu, \nu]$ is the set of joint measures on $M \times M$ with marginals $\mu$ and $\nu$. Defined as above, $W_p$ satisfies the properties of a metric. Furhermore, a minimizer in (4) is always achieved.

## 3  The Wasserstein metric for GPs

We will now study the Wasserstein metric with $p = 2$ between GPs. For GDs, this has been studied in [11, 14, 18, 22, 33].

From now on, assume that all GPs $f \sim \mathcal{GP}(m, k)$ are indexed over a compact $X \subset \mathbb{R}^n$ so that $\mathcal{H} := L^2(X)$ is separable. Furthermore, we assume $m \in L^2(X)$, $k \in L^2(X \times X)$, so that observations of $f$ live almost surely in $\mathcal{H}$. Let $f_1 \sim \mathcal{GP}(m_1, k_1)$ and $f_2 \sim \mathcal{GP}(m_2, k_2)$ be GPs with associated covariance operators $K_1$ and $K_2$, respectively. As the sample paths of $f_1$ and $f_2$ are in $\mathcal{H}$, they induce Gaussian measures $\mu_1, \mu_2 \in \mathcal{P}_2(\mathcal{H})$ on $\mathcal{H}$, as there is a 1-1 correspondence between GPs having sample paths almost surely on a $L^2(X)$ space and Gaussian measures on $L^2(X)$ [27].

The 2-Wasserstein metric between the Gaussian measures $\mu_1$, $\mu_2$ is given by [13]

$$W_2^2(\mu_1, \mu_2) = d_2^2(m_1, m_2) + \mathrm{Tr}\,(K_1 + K_2 - 2(K_1^{\frac{1}{2}} K_2 K_1^{\frac{1}{2}})^{\frac{1}{2}}), \tag{5}$$

where $d_2$ is the canonical metric on $L^2(X)$. Using this, we get the following definition

**Definition 1.** *Let $f_1$, $f_2$ be GPs as above, and the induced Gaussian measures of $f_1$ and $f_2$ be $\mu_1$ and $\mu_2$, respectively. Then, their squared 2-Wasserstein distance is given by*

$$W_2^2(f_1, f_2) := W_2^2(\mu_1, \mu_2) = d_2^2(m_1, m_2) + Tr\,(K_1 + K_2 - 2(K_1^{\frac{1}{2}} K_2 K_1^{\frac{1}{2}})^{\frac{1}{2}}) \,.$$

**Remark 2.** *Note that the case $m_1 = m_2 = 0$ defines a metric for the covariance operators $K_1, K_2$, as (5) shows that the space of GPs is isometric to the cartesian product of $L^2(X)$ and the covariance operators. We will denote this metric by $W_2^2(K_1, K_2)$. Furthermore, as GDs are just a subset of GPs, $W_2^2$ yields also the 2-Wasserstein metric between GDs studied in [11, 14, 18, 22, 33].*

**Barycenters of Gaussian processes.**    Next, we define and study barycenters of populations of GPs, in a similar fashion as the GD case in [1].

Given a population $\{\mu_i\}_{i=1}^N \subset \mathcal{P}_2(\mathcal{H})$ and weights $\{\xi_i \geq 0\}_{i=1}^N$ with $\sum_{i=1}^N \xi_i = 1$, and $\mathcal{H}$ a separable Hilbert space, the solution $\bar{\mu}$ of the problem

$$(\mathcal{P}) \quad \inf_{\mu \in \mathcal{P}_2(\mathcal{H})} \sum_{i=1}^N \xi_i W_2^2(\mu_i, \mu),$$

is the *barycenter* of the population $\{\mu_i\}_{i=1}^N$ with *barycentric coordinates* $\{\xi_i\}_{i=1}^N$. The barycenter for GPs is defined to be the barycenter of the associated Gaussian measures.

**Remark 3.** *The following theorems require the assumption that the barycenter is non-degenerate; it is still a conjecture that the barycenter of non-degenerate GPs is nondegenerate [20], but this holds in the finite-dimensional case of GDs.*

We now state the main theorem of this section, which follows from Prop. 5 and Prop. 6 below.

**Theorem 4.** *Let $\{f_i\}_{i=1}^N$ be a population of GPs with $f_i \sim \mathcal{GP}(m_i, K_i)$, then there exists a unique barycenter $\bar{f} \sim \mathcal{GP}(\bar{m}, \bar{K})$ with barycentric coordinates $(\xi_i)_{i=1}^N$. If $\bar{f}$ is non-degenerate, then $\bar{m}$ and $\bar{K}$ satisfy*

$$\bar{m} = \sum_{i=1}^N \xi_i m_i, \quad \sum_{i=1}^N \xi_i \left( \bar{K}^{\frac{1}{2}} K_i \bar{K}^{\frac{1}{2}} \right)^{\frac{1}{2}} = \bar{K}.$$

**Proposition 5.** *Let $\{\mu_i\}_{i=1}^N \subset \mathcal{P}_2(\mathcal{H})$ and $\bar{\mu}$ be a barycenter with barycentric coordinates $(\xi_i)_{i=1}^N$. Assume $\mu_i$ is regular for some $i$, then $\bar{\mu}$ is the unique minimizer of $(\mathcal{P})$.*

*Proof.* We first show that the map $\nu \mapsto W_2^2(\mu, \nu)$ is convex, and strictly convex if $\mu$ is a regular measure. To see this, let $\nu_i \in \mathcal{P}_2(\mathcal{H})$ and $\gamma_i^* \in \Gamma[\mu, \nu_i]$ be the optimal transport plans between $\mu$ and

$\nu_i$ for $i = 1, 2$, then $\lambda \gamma_1^* + (1 - \lambda) \gamma_2^* \in \Gamma[\mu, \lambda \nu_1 + (1 - \lambda) \nu_2]$ for $\lambda \in [0, 1]$. Therefore

$$
\begin{aligned}
W_2^2(\mu, \lambda \nu_1 + (1 - \lambda) \nu_2) &= \inf_{\gamma \in \Gamma[\mu, \lambda \nu_1 + (1-\lambda)\nu_2]} \int_{\mathcal{H} \times \mathcal{H}} d^2(x, y) d\gamma \\
&\leq \int_{\mathcal{H} \times \mathcal{H}} d^2(x, y) d(\lambda \gamma_1^* + (1 - \lambda) \gamma_2^*) \\
&= \lambda W_2^2(\mu, \nu_1) + (1 - \lambda) W_2^2(\mu, \nu_2),
\end{aligned}
$$

which gives convexity. Note that for $\lambda \in ]0, 1[$, the transport plan $\lambda \gamma_1^* + (1 - \lambda) \gamma_2^*$ splits mass. Therefore it cannot be the unique optimal plan between $\mu$ and $(1 - t)\nu_1 + t\nu_2$. As $\mu$ is regular, the optimal plan does not split mass, as it is induced by a map [3, Thm. 6.2.10], so we have strict convexity. From this follows the strict convexity of the object function in $(\mathcal{P})$. □

Next we characterize the barycenter, assuming it is non-degenerate, in the spirit of the finite-dinemsional case in [1, Thm. 6.1].

**Proposition 6.** *Let $\{f_i\}_{i=1}^N$ be a population of centered GPs, $f_i \sim \mathcal{GP}(0, K_i)$. Then $(\mathcal{P})$ has a unique solution $\bar{f} \sim \mathcal{GP}(0, \bar{K})$. If $\bar{f}$ is non-degenerate, then $\bar{K}$ is the unique bounded self-adjoint positive linear operator satisfying*

$$
\sum_{i=1}^N \xi_i \left( K^{\frac{1}{2}} K_i K^{\frac{1}{2}} \right)^{\frac{1}{2}} = K. \tag{6}
$$

*Proof.* Existence can be shown following the proof for the finite dimensional case [1, Prop. 4.2], which uses *multimarginal optimal transport*; this appears in the preprint [20, Cor. 9]. For the characterization, assume $\bar{f}$ to be non-degenerate, and let

$$
\mathrm{BC}(f) = \sum_{i=1}^N \xi_i W_2^2(f_i, f),
$$

be the barycentric expression, and assume that the minimizer $\bar{f}$ of BC is non-degenerate. Let $0 < \lambda_1, \lambda_2, ...$ be the eigenvalues of $\bar{K}$ with eigenfunctions $e_1, e_2, ....$ Then, by [10, Prop. 2.2.] the transport map between $\bar{f}$ and $f_k$ is given by

$$
T_k(x) = \sum_{i=1}^\infty \sum_{j=1}^\infty \frac{\langle x, e_j \rangle \langle (\bar{K}^{\frac{1}{2}} K_k \bar{K}^{\frac{1}{2}})^{\frac{1}{2}} e_j, e_i \rangle}{\lambda_i^{\frac{1}{2}} \lambda_j^{\frac{1}{2}}} e_i(x) \ . \tag{7}
$$

Using [6, Thm. 8.4.7], we can write the gradient of the barycentric expression. We furthermore know that the expression is strictly convex, thus the gradient at $\bar{f}$ equals zero if and only if $\bar{f}$ is the minimizer. Now let Id be the identity operator, then

$$
\nabla \mathrm{BC}(\bar{f}) = \sum_{i=1}^N (T_k - \mathrm{Id}) = 0,
$$

substituting in (7), we get

$$
\sum_{i=1}^N \xi_i \left( K^{\frac{1}{2}} K_i K^{\frac{1}{2}} \right)^{\frac{1}{2}} = K.
$$

□

*Proof of Theorem 4.* Use Prop. 6, the properties of a barycenter in a Hilbert space, and that the space of GPs is isometric to the cartesian product of $L^2(X)$ and the covariance operators. □

**Remark 7.** *For the practical computations of barycenters of GDs approximating GPs, to be discussed below, a fixed-point iteration scheme with a guarantee of convergence exists [4, Thm. 4.2].*

**Convergence properties.** Now, we show that the 2-Wasserstein metric for GPs can be approximated arbitrarily well by the 2-Wasserstein metric for GDs. This is important, as in real-life we observe finite-dimensional representations of the covariance operators.

Let $\{e_i\}_{i=1}^{\infty}$ be an orthonormal basis for $L^2(X)$. Then we define the GDs given by restrictions $m_{in}$ and $K_{in}$ of $m_i$ and $K_i$, $i = 1, 2$, on $V_n = \text{span}(e_1, ..., e_n)$ by

$$m_{in}(x) = \sum_{k=1}^{n} \langle m_i, e_k \rangle e_k(x), \ K_{in}\phi = \sum_{k=1}^{n} \langle \phi, e_k \rangle K_i e_k, \ \forall \phi \in V_n, \ \forall x \in X , \qquad (8)$$

and prove the following:

**Theorem 8.** *The 2-Wasserstein metric between GDs on finite samples converges to the Wasserstein metric between GPs, that is, if $f_{in} \sim \mathcal{N}(m_{in}, K_{in})$, $f_i \sim \mathcal{GP}(m_i, K_i)$ for $i = 1, 2$, then*

$$\lim_{n \to \infty} W_2^2(f_{1n}, f_{2n}) = W_2^2(f_1, f_2).$$

*By the same argument, it also follows that $W_2^2(\cdot, \cdot)$ is continuous in both arguments in operator norm topology.*

*Proof.* $K_{in} \to K_i$ in operator norm as $n \to \infty$. Because taking a sum, product and square-root of operators are all continuous with respect to the operator norm, it follows that

$$K_{1n} + K_{2n} - 2(K_{1n}^{\frac{1}{2}} K_{2n} K_{1n}^{\frac{1}{2}})^{\frac{1}{2}} \to K_1 + K_2 - 2(K_1^{\frac{1}{2}} K_2 K_1^{\frac{1}{2}})^{\frac{1}{2}}.$$

Note that for any sequence $A_n \to A$ with convergence in operator norm, we have

$$|\text{Tr}\, A - \text{Tr}\, A_n| \leq \sum_{k=1}^{\infty} |\langle (A - A_n)e_k, e_k \rangle| \overset{\text{Cauchy-Schwarz}}{\leq} \sum_{k=1}^{\infty} \|(A - A_n)e_k\|_{L^2} \overset{\text{MCT}}{\to} 0 , \quad (9)$$

as $\lim_{n \to \infty} \sup_{v \in L_\omega^2(X)} \|(A - A_n)v\|_{L^2} = 0$ due to the convergence in operator norm. Here MCT stands for the monotone convergence theorem. Thus we have

$$W_2^2(f_{1n}, f_{2n}) = d_2^2(m_{1n}, m_{2n}) + \text{Tr}\, (K_{1n} + K_{2n} - 2(K_{1n}^{\frac{1}{2}} K_{2n} K_{1n}^{\frac{1}{2}})^{\frac{1}{2}})$$
$$\overset{n \to \infty}{\to} d_2^2(m_1, m_2) + \text{Tr}\, (K_1 + K_2 - 2(K_1^{\frac{1}{2}} K_2 K_1^{\frac{1}{2}})^{\frac{1}{2}})$$
$$= W_2^2(f_1, f_2).$$

$\square$

The importance of Proposition 8 is that it justifies computations of distances using finite representations of GPs as approximations for the infinite-dimensional case.

Next, assuming the barycenter is non-degenerate, we show that we can also approximate the barycenter of a population of GPs by computing the barycenters of populations of GDs converging to these GPs. In the degenerate case, see [20, Thm. 11].

**Theorem 9.** *Assuming the barycenter of a population of GPs is non-degenerate, then it varies continuously, that is, the map $(f_1, ..., f_N) \mapsto \bar{f}$ is continuous in the operator norm. Especially, this implies that the barycenter $\bar{f}_n$ of the finite-dimensional restrictions $\{f_{in}\}_{i=1}^{N}$ converges to $\bar{f}$.*

First, we show that if $f_i \sim \mathcal{GP}(m_i, K_i)$ and $\bar{f} = \mathcal{GP}(\bar{m}, \bar{K})$, then that the map $(K_1, ..., K_N) \mapsto \bar{K}$ is continuous. Continuity of $(m_1, ..., m_N) \mapsto \bar{m}$ is clear.

Let $K$ be a covariance operator, denote its maximal eigenvalue by $\lambda_{\max}(K)$. Note that this map is well-defined, as $K$ is also bounded, normal operator, thus $\lambda_{\max}(K) = \|K\|_{op} < \infty$ holds. Now let $\mathbf{a} = (K_1, ..., K_N)$ be a population of covariance operators, denote $i^{\text{th}}$ as $\mathbf{a}(i) = K_i$, then define the continuous function $\beta$ and correspondence (a set valued map) $\Phi$ as follows

$$\beta : \mathbf{a} \mapsto \left( \sum_{i=1}^{N} \xi_i \sqrt{\lambda_{max}(\mathbf{a}(i))} \right)^2 , \ \Phi : \mathbf{a} \mapsto K_{\beta(\mathbf{a})} = \{K \in \text{HS}(\mathcal{H}) \mid \beta(\mathbf{a})I \geq K \geq 0\}.$$

Then the fixed point of (6) can be found in $\Phi(\mathbf{a})$, as the map

$$F(K) = \sum_{i=1}^{N} \xi_i \left( K^{\frac{1}{2}} K_i K^{\frac{1}{2}} \right)^{\frac{1}{2}},$$

is a compact operator, $\Phi(\mathbf{a})$ is bounded, and so the closure of $F(\Phi(\mathbf{a}))$ is compact. Furthermore, do note that $F$ is a map from $\Phi(\mathbf{a})$ to itself, so by Schauder's fixed point theorem, there exists a fixed point.

Now, we want to show that this correspondence is continuous in order to put the Maximum theorem to use. A correspondence $\Phi : A \to B$ is *upper hemi-continuous* at $a \in A$, if all convergent sequences $(a_n) \in A$, $(b_n) \in \Phi(a_n)$ satisfy $\lim_{n\to\infty} b_n = b$, $\lim_{n\to\infty} a_n = a$ and $b \in \Phi(a)$. The correspondence is *lower hemi-continuous* at $a \in A$, if for all convergent sequences $a_n \to a$ in $A$ and any $b \in \Phi(a)$, there is a subsequence $a_{n_k}$, so that we have a sequence $b_k \in \Phi(a_{n_k})$ which satisfies $b_k \to b$. If the correspondence is both upper and lower hemi-continuous, we say that it is *continuous*. For more about the Maximum theorem and hemi-continuity, see [2].

**Lemma 10.** *The correspondence* $\Phi : \mathbf{a} \mapsto K_{\beta(\mathbf{a})}$ *is continuous as correspondence.*

*Proof.* First, we show the correspondence is lower hemi-continuous. Let $(\mathbf{a}_n)_{n=1}^{\infty}$ be a sequence of populations of covariance operators of size $N$, that converges $\mathbf{a}_n \to \mathbf{a}$. Use the shorthand notation $\beta_n := \beta(\mathbf{a}_n)$, then $\beta_n \to \beta_\infty := \beta(\mathbf{a})$, and let $\mathbf{b} \in \Phi(\mathbf{a}) = K_{\beta_\infty}$.

Pick subsequence $(\mathbf{a}_{n_k})_{k=1}^{\infty}$ so that $(\beta_{n_k})_{k=1}^{\infty}$ is increasing or decreasing. If it was decreasing, then $K_{\beta_\infty} \subseteq K_{\beta_{n_k}}$ for every $n_k$. Thus the proof would be finished by choosing $\mathbf{b}_k = \mathbf{b}$ for every $k$. Hence assume the sequence is increasing, so that $K_{\beta_{n_k}} \subseteq K_{\beta_{n_{k+1}}}$. Now let $\gamma(t) = (1-t)\mathbf{b}_1 + t\mathbf{b}$, where $\mathbf{b}_1 \in K_{\beta_1}$, and let $t_{n_k}$ be the solution to $(1-t)\beta_1 + t\beta_\infty = \beta_{n_k}$, then $\mathbf{b}_k := \gamma(t_{n_k}) \in K_{\beta_{n_k}}$ and $\mathbf{b}_k \to \mathbf{b}$.

For upper hemicontinuity, assume that $\mathbf{a}_n \to \mathbf{a}$, $\mathbf{b}_n \in K_{\beta_n}$ and that $\mathbf{b}_n \to \mathbf{b}$. Then using the definition of $\Phi$, we get the positive sequence $\langle(\beta_n I - \mathbf{b}_n)x, x\rangle \geq 0$ indexed by $n$, then by continuity and the positivity of this sequence it follows that

$$0 \leq \lim_{n\to\infty} \langle(\beta_n I - \mathbf{b}_n)x, x\rangle = \langle(\beta_\infty I - \mathbf{b})x, x\rangle.$$

One can check the criterion $\mathbf{b} \geq 0$ similarly, and so we are done. $\qquad\square$

*Proof of Theorem 9.* Now let $\mathbf{a} = (K_1, ..., K_n)$, $\mathbf{f}(K, \mathbf{a}) := \sum_{i=1}^{N} \xi_i W_2^2(K, K_i)$ and $F(K) := \sum_{i=1}^{N} \xi_i (K^{\frac{1}{2}} K_i K^{\frac{1}{2}})^{\frac{1}{2}}$, then the unique minimizer $\bar{K}$ of $\mathbf{f}$ is the fixed point of $F$. Furthermore, the closure $\mathrm{cl}(F(K_{\beta(\mathbf{a})}))$ is compact, $\mathbf{a} \mapsto \mathrm{cl}(F(K_{\beta(\mathbf{a})}))$ is a continuous correspondence as the closure of composition of two continuous correspondence. Additionally, we know that $\bar{K} \in \mathrm{cl}(F(K_{\beta(a)}))$, so applying the maximum theorem, we have shown that the barycenter of a population of covariance operators varies continuously, i.e. the map $(K_1, ..., K_N) \mapsto \bar{K}$ is continuous, finishing the proof. $\quad\square$

## 4 Experiments

We illustrate the utility of the Wasserstein metric in two different applications: Processing of uncertain white-matter tracts estimated from DWI, and analysis of climate development via temperature curve GPs.

**Experimental setup.** The white-matter tract GPs are estimated for a single subject from the Human Connectome Project [15, 32, 35], using probabilistic shortest-path tractography [17]. See the supplementary material for details on the data and its preprocessing. From daily minimum temperatures measured at a set of 30 randomly sampled Russian metereological stations [9, 34], GP regression was used to estimate a GP temperature curve per year and station for the period $1940 - 2009$ using maximum likelihood parameters. All code for computing Wasserstein distances and barycenters was implemented in MATLAB and ran on a laptop with $2,7$ GHz Intel Core i5 processor and 8 GB 1867 MHz DDR3 memory. On the temperature GP curves (represented by 50 samples), the average runtime of the 2-Wasserstein distance computation was $0.048 \pm 0.014$ seconds (estimated from 1000 pairwise distance computations), and the average runtime of the 2-Wasserstein barycenter of a sample of size 10 was $0.69 \pm 0.11$ seconds (estimated from 200 samples).

**White-matter tract processing.** The *inferior longitudinal fasiculus* is a white-matter bundle which splits into two separate bundles. Fig. 3 (top) shows the results of agglomerative hierarchical clustering of the GP tracts using average Wasserstein distance. The per-cluster Wasserstein barycenter can be used to represent the tracts; its overlap with the individual GP mean curves is shown in Fig. 3 (bottom).

The individual GP tracts are visualized via their mean curves, but they are in fact a population of GPs. To confirm that the two clusters are indeed different also when the covariance function is taken into account, we perform a permutation test for difference between per-cluster Wasserstein barycenters, and already with 50 permutations we observe a $p$-value of $p = 0.0196$, confirming that the two clusters are significantly different at a $5\%$ significance level.

**Quantifying climate change.** Using the Wasserstein barycenters we perform nonparametric kernel regression to visualize how yearly temperature curves evolve with time, based on the Russian yearly temperature GPs. Fig. 4 shows snapshots from this evolution, and a continuous movie version `climate.avi` is found in the supplementary material. The regressed evolution indicates an increase in overall temperature as we reach the final year 2009. To quantify this observation, we perform a permutation test using the Wasserstein distance between population Wasserstein barycenters to compare the final 10 years 2000-2009 with the years 1940-1999. Using 50 permutations we obtain a $p$-value of 0.0392, giving significant difference in temperature curves at a $95\%$ confidence level.

**Significance.** Note that the state-of-the-art in tract analysis as well as in functional data analysis would be to ignore the covariance of the estimated curves and treat the mean curves as observations. We contribute a framework to incorporate the uncertainty into the population analysis – but why would we want to retain uncertainty? In the white-matter tracts, the GP covariance represents spatial uncertainty in the estimated curve trajectory. The individual GPs represent connections between different endpoints. Thus, they do not represent observations of the exact same trajectory, but rather of distinct, nearby trajectories. It is common in diffusion MRI to represent such sets of estimated trajectories by a few prototype trajectories for visualization and comparative analysis; we obtain prototypes through the Wasserstein barycenter. To correctly interpret the spatial uncertainty, e.g. for a brain surgeon [8], it is crucial that the covariance of the prototype GP represents the covariances of the individual GPs, and not smaller. If you wanted to reduce uncertainty by increasing sample size, you would need more images, not more curves – because the noise is in the image. But more images are not usually available. In the climate data, the GP covariance models natural temperature variation, *not* measurement noise. Increasing the sample size decreases the error of the temperature distribution, but should not decrease this natural variation (i.e. the covariance).

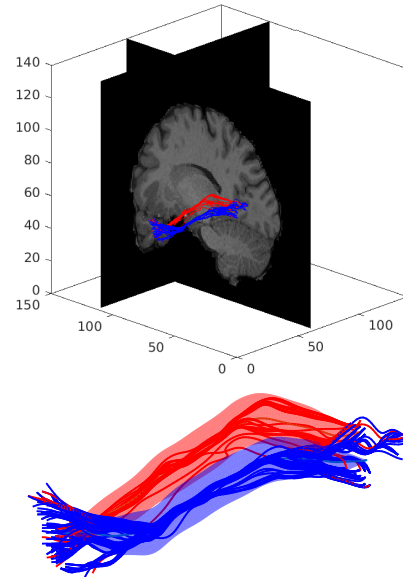

Figure 3: **Top:** The mean functions of the individual GPs, colored by cluster membership, in the context of the corresponding T1-weighted MRI slices. **Bottom:** The tract GP mean functions and the cluster mean GPs with 95% confidence bounds.

## 5  Discussion and future work

We have shown that the Wasserstein metric for GPs is both theoretically and computationally well-founded for statistics on GPs: It defines unique barycenters, and allows efficient computations through finite-dimensional representations. We have illustrated its use in two different applications: Processing of uncertain estimates of white-matter trajectories in the brain, and analysis of climate development via GP representations of temperature curves. We have seen that the metric itself is discriminative for clustering and permutation testing, and we have seen how the GP barycenters allow truthful interpretation of uncertainty in the white matter tracts and of variation in the temperature curves.

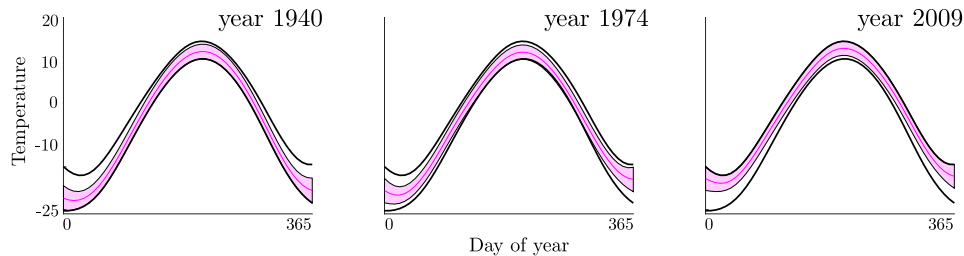

Figure 4: Snapshots from the kernel regression giving yearly temperature curves 1940-2009. We observe an apparent temperature increase which is confirmed by the permutation test.

Future work includes more complex learning algorithms, starting with preprocessing tools such as PCA [31], and moving on to supervised predictive models. This includes a better understanding of the potentially Riemannian structure of the infinite-dimensional Wasserstein space, which would enable us to draw on existing results for learning with manifold-valued data [21].

The Wasserstein distance allows the inherent uncertainty in the estimated GP data points to be appropriately accounted for in every step of the analysis, giving truthful analysis and subsequent interpretation. This is particularly important in applications where uncertainty or variation is crucial: Variation in temperature is an important feature in climate change, and while estimated white-matter trajectories are known to be unreliable, they are used in surgical planning, making uncertainty about their trajectories a highly relevant parameter.

# 6 Acknowledgements

This research was supported by Centre for Stochastic Geometry and Advanced Bioimaging, funded by a grant from the Villum Foundation. Data were provided [in part] by the Human Connectome Project, WU-Minn Consortium (Principal Investigators: David Van Essen and Kamil Ugurbil; 1U54MH091657) funded by the 16 NIH Institutes and Centers that support the NIH Blueprint for Neuroscience Research; and by the McDonnell Center for Systems Neuroscience at Washington University. The authors would also like to thank Mads Nielsen for valuable discussions and supervision. Finally, the authors would like to thank Victor Panaretos for valuable discussions and, in particular, for pointing out an error in an earlier version of the manuscript.

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
