[Supplementary Material]

# Supplementary material for
# Learning from uncertain curves:
# The $2$-Wasserstein metric for Gaussian Processes

## 1   Content

The supplementary material consists of the movie file `climate.avi` as well as the appendices listed below.

## A   Details of data processing

### A.1   The uncertain white-matter tract trajectories

The white-matter tract example is carried out on subject 103414 from the Q3 release of the Human Connectome Project [2–4]. The pre-processed HCP diffusion data contains 270 diffusion directions distributed equally over 3 shells with $b$-values $= 1000, 2000$ and $3000 \text{ s/mm}^2$. We perform probabilistic shortest-path tractography [5], which returns uncertain tracts in the form of GPs by estimating uncertain geodesics in an uncertain Riemannian manifold as described in [5]. The produced GPs are indexed over $X = [0, 1]$, have continuous covariance functions and $L^2$ integrable means, thus allowing us to apply the developed framework.

### A.2   The GP temperature curves

The temperature curve GPs were estimated from a publicly available dataset of temperature measurements from Russian metereological stations [6, 7]. We used data from the years $1940 - 2009$, and 30 stations were randomly sampled among those that covered all these years. For each year, and each station, a GP was fitted to the measurements of minimal daily temperature, using a Gaussian covariance function. The maximum likelihood scale and bandwidth parameters of the covariance function were estimated using grid search. Years where more than 20 daily measurements were missing, were discarded from the dataset.

## B   Details of kernel regression

Given two metric spaces $(X, d_X)$ and $(Y, d_Y)$, where the barycenter

$$\text{argmin}_{y \in Y} \sum_{n=1}^{N} \xi_n d_Y^2(y, y_n)$$

is well defined, we can define a nonparametric *kernel regression* model as follows:

Given a sample of pairs $(x_1, y_1), \ldots, (x_N, y_N) \in X \times Y$ of free variables $x \in X$ and dependent variables $y \in Y$, we predict $Y$-valued observations $y_0 = h(x_0)$ given new observations $x_0$ of $x$, as

follows:

$$y_0 = \operatorname{argmin}_{y \in Y} \sum_{n=1}^{N} k(x_n, x_0) d_Y^2(y, y_n)$$

where $k$ is some kernel function. In our case, we use the Gaussian kernel

$$k(x_n, x_0) = C \cdot e^{-\frac{1}{2\sigma^2} d_X^2(x_n, x_0)},$$

where $C$ is set to ensure that $\sum_{n=1}^{N} k(x_n, x_0) = 1$.

## C  Details of permutation test

Given a metric space $(X, d_X)$ where the barycenter

$$\mu = \operatorname{argmin}_{y \in Y} \sum_{n=1}^{N} \xi_n d_Y^2(y, y_n)$$

is well defined, we can also define nonparametric hypothesis testing through permutation tests. Assume that $A = \{a_i\}_{i=1}^{N_1}$ and $B = \{b_j\}_{j=1}^{N_2}$ are two finite, sampled datasets in $X$; now we define the test statistic $T(A, B) = d(\mu_A, \mu_B)$, where $\mu_A$ and $\mu_B$ are barycenters of the sets $A$ and $B$, respectively. Under the null hypothesis the samples $A$ and $B$ are drawn from the same distribution on $\mathcal{T}$, and randomly permuting the elements of $A$ and $B$ should not affect the value of $T$.

Join the sets in a larger dataset $S = A \cup B$ and consider partitions of $X$ into datasets of size $N_1$ and $N_2$. Due to the size of $X$ we cannot usually check all possible permutations, but compute the test statistics $T_m = d(\mu_{A_m}, \mu_{B_m})$, $m = 1, \dots, M$, for $M$ different random partitions $X = A_m \cup B_m$, with $|A_m| = N_1$ and $|B_m| = N_2$. Comparing the $T_m$ to the original statistic value $T_0 = d(\hat{\mu}_A, \hat{\mu}_B)$ we obtain a $p$-value approximating the probability of observing $T_0$ under the null hypothesis:

$$p = \frac{1 + \sum_{T_m \geq T_0, m \in \{1, \dots, M\}} 1}{M + 1},$$

where the additional 1 is added to avoid $p = 0$, which is impossible in the limit where all permutations are tested [1].