[Reviews · NeurIPS 2017]

Reviewer 1



The authors introduce a new framework for the incorporation of uncertainty in population analysis. They study populations of Gaussian Processes and define an approximation of the Wasserstein metric, showing that the barycenter exists and is unique. All the findings are well justified. The proposed method is validated by experimental results. Two interesting applications are presented: (1) the processing of white-matter trajectories in the brain, from diffusion weighted imaging; (2) the analysis of climate evolution, from data collected by Russian metereologiacal stations. The authors are able to succesfully estimate the uncertainty in the evolution of the learned curves. The experimental set up is well described and the computational cost of the method discussed. In the final discussion, it is interesting to see how the authors connect their work to the recent findings in [37] and [38], envisioning the potential for future work. Could you add few more details about your vision? Quality This paper is pleasant to read. The results presented are well supported by theoretical analysis as well as by experimental results. Clarity The paper is clear, well organised and tidy. All the figures are well presented. The experimental setup is well described and the author state that they will release the code upon publication. Originality The theoretical framework introduced in the paper is novel. Significance The method presented in the paper can be defined significant since the authors provide interpretable results for critical application domains. Moreover, this method is general and can be applied to different problems.

Reviewer 2



This paper proposes an optimal transport-based metric on Gaussian processes. The formula itself (equation 5) is a direct consequence of the 2-Wasserstein metric on Gaussian measures, which is well-studied. The authors propose this formula and derive a fixed-point iteration for computing the barycenter of a set of Gaussian processes under this metric, whose convergence appears to be a consequence of [26] (a bit more detail in this part of the discussion would be appreciated). The authors prove nice mathematical properties for the barycenter/transport problems in this metric in a sequence of theorems (continuity wrt operator norm, convergence of distance as basis size increases), and some experiments in diffusion tractography and climate change. My main question after reading this document is whether there is a nice intuition for the GP transportation metric. In particular, Wasserstein metrics are derived using a linear program (or convex optimization problem for measures, in the continuum case). The paper submission jumps directly to a formula (5) without any optimization-based interpretation of optimal transport for GPs. What quantity are we transporting? How should I think about this metric specifically in the context of GPs? The theoretical properties of this metric are nice and not too difficult to prove as consequences of properties of optimal transport between Gaussian measures. The experiments are less convincing --- in what circumstances does using this geometrization of the set of GPs really help? The discussion in this section just sort of indicates that the authors computed the Wasserstein barycenter/distance in GP space, but doesn't indicate strongly if/why it provided any benefit for the climate change or tractography applications over existing models. Other comments: * l.62: Capitalize Procrustes * It might be worth noting that optimal transport between Gaussian measures is also known as the "Bures metric." See e.g. "On the Geometry of Covariance Matrices" (Ning, Jiang, Georgiou). * Not sure if it's relevant, but it might be worth noting any relationship to the method in "Gromov-Wasserstein Averaging of Kernel and Distance Matrices" near eq (2). * Is it possible to provide a transportation distance on GPs for other values of p? * The notation in the equation after l. 130 caught me by surprise --- in particular, the xi's are input to the problem rather than optimization variables so I might recommend moving the 0 < xi_i and sum_i xi_i = 1 to be inline with the following paragraph. Also note that there's a typo under the inf. * What do you mean by "indexed" on l.114? * l.178: GDs * Please expand section 4 to include more detailed **interpretation** of the results rather than just telling us how they were generated. Was the new distance on GPs worth it? What improved? This wasn't clear to me from the text. * Note for future submissions that section 6 should be omitted for anonymity... * Several typos in the references (capitalization, missing info in BibTeX) should be fixed.

Reviewer 3



The paper extends the Wasserstein (earth-mover's) metric from finite-dimensional Gaussians to GPs, defining a geometry on spaces of GPs and in particular the notion of a barycenter, or 'average' GP. It establishes that 2-Wasserstein distances, and barycenters, between GPs can be computed as the limit of equivalent computations on finite Gaussians. These results are applied to analysis of uncertain white-matter tract data and annual temperature curves from multiple stations. The writing is serviceable and relevant background is presented clearly. I found no errors in the results themselves, though did not check all of them thoroughly. My main concerns are with motivation and significance. I agree that it is in some sense natural to extend the Wasserstein geometry on Gaussians to GPs, and it's reassuring to see that this can be done and (as I understand the results of the paper) basically nothing surprising happens -- this is definitely a contribution. But there is not much motivation for why this is a useful thing to do. The experiments in the paper are (as is standard for theory papers) fairly contrived, and the conclusions they arrive at (the earth is getting warmer) could likely be gained by many other forms of analysis --- what makes the Wasserstein machinery particularly meaningful here? For example: from a hierarchical Bayesian perspective it would seem reasonable to model the climate data using a latent, unobserved global temperature curve f_global ~ GP(m_global, k_global), and then assume that each station i observes that curve with additional local GP noise, f_i ~ GP(f_global + mi, ki). Then the latent global temperature curve could be analytically computed through simple Gaussian conditioning, would be interpretable in terms of this simple generative model, and would (correctly) become more certain as we gathered evidence from additional stations. By contrast I don't understand how to statistically interpret the barycenter of independently estimated GPs, or what meaning to assign to its uncertainty (am I correct in understanding that uncertainty would *not* decrease given additional observations? what are the circumstances in which this is desirable behavior?). Perhaps other reviewers with a more theoretical bent will find this work exciting. But in the absence of any really surprising results or novel theoretical tools, I would like to see evidence that the machinery it builds is well-motivated and has real advantages for practical data analysis -- can it reach conclusions more efficiently or uncover structure more effectively than existing techniques? As currently written the paper doesn't cross my personal NIPS significance bar. misc notes: - Fig 1: the 'naive mean' column claims to average the pointwise standard deviations, but the stddevs shown are smaller than *any* of the example curves: how can this be true? Also why is averaging pointwise stddevs, rather than (co)variances or precisions, the relevant comparison? - I wonder if the Wasserstein metric could be useful for variational fitting of sparse/approximate GP models? Currently this is mostly done by minimizing KL divergence (as in Matthews et al. 2015, https://arxiv.org/abs/1504.07027) but one could imagine minimizing the Wasserstein distance between a target GP and a fast approximating GP. If that turned out to yield a practical method with statistical or computational advantages over KL divergence for fitting large-scale GPs, that'd be a noteworthy contribution. 130-131: missing close-paren in inf_\mu\inP2 (H 152: 'whe' -> 'we' 155: 'This set is a convex' 170: why is 'Covariance' capitalized? 178: 'GDss' -> 'GDs'?